# Effects of Wooden Breast Myopathy on Meat Quality Characteristics of Broiler Pectoralis Major Muscle and Its Changes with Intramuscular Connective Tissue

**DOI:** 10.3390/foods13040507

**Published:** 2024-02-06

**Authors:** Tianjiao Bian, Tong Xing, Xue Zhao, Xinglian Xu

**Affiliations:** 1State Key Laboratory of Meat Quality Control and Cultured Meat Development, Jiangsu Collaborative Innovation Center of Meat Production and Processing, Quality and Safety Control, Nanjing Agricultural University, Nanjing 210095, China; 2Key Laboratory of Animal Origin Food Production and Safety Guarantee of Jiangsu Province, College of Animal Science and Technology, Nanjing Agricultural University, Nanjing 210095, China

**Keywords:** broiler, wooden breast, meat quality, collagen, intramuscular connective tissue

## Abstract

This study aimed to investigate the effect of wooden breast (WB) myopathy on chemical composition, meat quality attributes and physiochemical characteristics of intramuscular connective tissue (IMCT) of broiler pectoralis major (PM) muscle. Thirty-six fillets were classified into varying degrees of WB condition, including normal, moderate and severe. Results show that WB myopathy altered the collagen profile in PM muscle by increasing total collagen content and decreasing collagen solubility. The composition of macromolecules in IMCT, including hydroxylysyl pyridoxine cross-linking, decorin and glycosaminoglycans, were increased with the severity of WB myopathy. Differential scanning calorimetry analysis indicated higher denaturation temperatures and lower denaturation enthalpy of IMCT for WB. Secondary structures of α-helix and β-sheet in the IMCT of WB were changed to β-turn and random coil. In addition, chemical composition and meat quality attributes showed a correlation with collagen profile and IMCT characteristics. Overall, this study emphasizes the effect of WB myopathy on IMCT and their contributions to meat quality variation.

## 1. Introduction

Chicken meat is globally preferred due to its healthy nutritional profile, sensory properties, and compatibility with diverse religious and cultural customs. The poultry industry has made significant progress in selecting broiler chickens for accelerated growth and enhanced muscle yield to meet with the increasing demand of poultry meat. However, this has been accompanied by increased incidences of breast muscle abnormalities over the past decade [1]. Wooden breast (WB) myopathy, one of the most serious muscular defects, affects the global poultry industry and leads to remarkable economic losses. Macroscopically, WB exhibits pale expansive areas of substantial hardness, often accompanied with white striping, minor hemorrhages and clear viscous fluid [2]. WB fillets can be typically identified through palpation evaluation and are usually classified into several grades based on the degree of hardness [3].

The histopathologic features of WB are polyphasic muscle fiber myodegeneration with partial regeneration, infiltration of inflammatory cells, and various amounts of interstitial connective tissue deposition [2]. Although the precise etiology of this growth-related myopathy remains not fully understood, accumulating evidence has shown that localized tissue hypoxia, oxidative stress, mitochondrial dysfunction and glucolipotoxicity are involved in the development of WB [1,4,5]. Among the histological features, fibrosis featured by the extensive deposition of intramuscular connective tissue (IMCT) is the most typical lesion of WB. The excessive deposition of extracellular matrix (ECM) and the cross-linking of collagen fibrils can lead to compromised muscle contraction and the increase of chicken breast muscle stiffness, which is the rationale behind the vernacular name ‘wooden breast’ [6,7]. In addition, the excessive collagen tissue deposition certainly leads to a decrease in nutritional value and compromised meat quality traits [8,9].

IMCT, including epimysium, endomysium, and perimysium, maintains the structural integrity of muscle fiber. IMCT principally consists of ECM macromolecules, including collagens, proteoglycans (PGs) and glycoproteins [10]. The composition, structure, and physiochemical properties of IMCT have been shown to significantly contribute to muscle development and meat quality formation [11]. Collagen, the most abundant IMCT component, plays a crucial role in determining the texture of raw and cooked meat [12]. Collagen obtains its physiochemical stability through intramolecular and intermolecular cross-linking, and mature cross-links cause collagenous fibers to continuously form a three-dimensional structure to increase muscle toughness [13]. Meanwhile, cross-links increase the thermal stability of collagen molecules, thereby reducing meat tenderness [14]. As another principal component of IMCT, PGs link collagen fibrils and regulate the size and arrangement of collagen matrix, while also exerting important roles during meat quality formation [15]. Despite extensive studies assessing the impact of WB abnormality on meat quality traits, technological properties and even collagen profiles [1,16,17], the exact compositional and physiochemical changes that occur in IMCT and their contributions to the meat quality of broiler fillets with WB still require further research.

This study aimed to explore alterations in the constitution or composition of macromolecules in IMCT as well as the physiochemical properties of IMCT, especially collagen, and to evaluate their contributions to the chemical composition, meat quality and textural profiles of WB fillets with different severity.

## 2. Materials and Methods

### 2.1. Broiler Breast Fillets Collection

Broiler breast fillets (pectoralis major muscles) from the left side were collected from the same commercial broiler line of Arbor Acres at a commercial processing plant (Yike Inc., Suqian, Jiangsu, China) on the day of slaughter. The slaughter process adhered to standard industrial procedures, which includes electrical stunning, bleeding, scalding, evisceration, chilling and deboning. Fillets were randomly selected from a deboning line (at approximately 2 h postmortem) and subjected to tactile evaluation for the categorization of WB condition as described by Tijare et al. [3] and including three groups: normal, which comprises fillets that were flexible throughout; moderate, which comprises fillets that were hard throughout but flexible in the mid-to-caudal region; and severe, which comprises fillets that were extremely hard and rigid throughout, from the cranial region to the caudal tip. Finally, a total of 36 fillets (12 fillets per category) were identified and stored at 4 °C for further assessment.

### 2.2. Histological Evaluation

Three muscle tissues were randomly selected from each group, fixed in 4% paraformaldehyde and embedded in paraffin blocks. Subsequently, cross-sections were cut into 8 μm thickness and then subjected to hematoxylin and eosin (H&E) staining, Masson Trichrome staining, and Sirius red staining according to previous studies [7,18] Using a light microscope, images were acquired at consistent magnification and settings. Slides were examined under a bright-field microscope with standardized illuminator and condenser settings, and images were acquired at ×100 magnification (Axio Scope.A1, Carl Zeiss, Oberkochen, Germany).

### 2.3. Chemical Composition Analysis

The moisture, protein, fat, and ash contents of broiler breast meat were measured in accordance with the standard procedures outlined by the Association of Official Analytical Chemists [19].

### 2.4. Meat Quality Measurement

At 24 h postmortem, breast fillets underwent meat quality assessment as outlined in previous studies [20]. Meat color was determined thrice at the cranial section on the dorsal surface of each fillet using a Minolta Chroma Meter (Konica Minolta Company, Tokyo, Japan). Muscle pH was measured in triplicate by inserting a probe electrode into the fillet’s cranial region using a HI9125 portable pH meter (Hanna Instruments, Padova, Italy).

A specially shaped muscle tissue (5 cm × 5 cm × 2 cm) was cut along with the muscle fiber from the cranial section of each fillet, weighted and placed into a cooking bag, and then cooked in a water bath at 80 °C until the central temperature reached 70 °C [21]. After cooling to room temperature in running water, meat samples were reweighted to calculate cooking loss (%). The cooked meat samples were cut into two strips (3 cm × 1 cm × 1 cm). Each strip underwent two perpendicular shears along the axis of the muscle fibers using a digital meat tenderness analyzer, with the maximum force (N) indicating the shear force value [20].

### 2.5. Texture Profile Analyses

Meat samples of two cylinders (diameter: 20 mm, height: 20 mm), perpendicular to the direction of muscle fiber, were cut for texture profile measurement with a texture analyzer (XT Plus, Stable Micro Systems Ltd., Godalming, UK) according to Dondero et al. [22]. The samples were subjected to the following settings: two compression cycles with 50% compression, a pre-test rate of 5 mm/s, a test rate of 5 mm/s, a post-test rate of 10 mm/s, a trigger force of 5 g, and a test time of 1 s.

### 2.6. NMR Transverse Relaxation (T_2_) Measurements

The low-field NMR spin-spin relaxation measurements were conducted using a Niumag Pulsed NMR analyzer (PQ001; Niumag Corporation, Shanghai, China) following the methodology outlined by Xing et al. [23]. A regular-shaped muscle sample (1.5 cm × 1 cm × 1 cm) was taken from each fillet for testing. The analyzer was set at a resonance frequency of 22.4 MHz and maintained at a temperature of 32 °C. The transverse relaxation time (T_2_) was determined by utilizing the Carr–Purcell–Meiboom–Gill (CPMG) sequence with a τ-value of 300 μs. Data from 3200 echoes were collected through 32 scan repetitions and fitted using the program Multi Exp Inv Analysis.

### 2.7. Collagen Profiles Measurements

For total collagen content analysis, approximately 4.0 g of samples were subjected to hydrolysis in 30 mL of sulfuric acid (3 M) for 16 h at 105 °C. The measurement of hydroxyproline content was conducted following the method outlined by Latorre et al. [24]. The content of total collagen was estimated from the hydroxyproline content using a conversion ratio of 7.25 and subsequently reported as a percentage.

For the determination of heat soluble collagen content, approximately 4.0 g of samples were homogenized in 8 mL of 1/4 Ringer’s solution by Nishimura et al. [25]. The homogenates underwent heating for 1 h at 77 °C, followed by centrifugation for 1200 s at 4000 g. The amount of soluble collagen in the combined supernatant was detected utilizing the aforementioned procedures. Collagen solubility is defined as the ratio of heat-soluble collagen to total amount of collagen.

### 2.8. Determination of Hydroxylysyl Pyridoxine, Glycosaminoglycan and Decorin

Frozen muscle tissues weighing approximately 0.5 g were diced and combined with 4.5 mL of PBS (0.01 M, pH = 7.4, 4 °C). The mixture underwent homogenization on ice at 10,000 rpm for 1 min, and then centrifugated at 5000× *g* for 10 min to collect the supernatant. Hydroxylysine pyridoxine (HP) content was determined using an enzyme-linked immunosorbent assay (ELISA) kit outlined by Du et al. [26], with HP levels expressed as μg/g muscle. Similarly, glycosaminoglycan and decorin levels were assessed using corresponding ELISA kits (Maibo Biotechnology Co., Ltd., Nanjing, China) and expressed as ng/g muscle, following the manufacturers’ instructions.

### 2.9. Extraction of Intramuscular Connective Tissue

The extraction of IMCT was conducted following the method outlined by Wu [27], with minor adjustments. Muscle tissues were divided into uniform pieces (1 cm× 1 cm × 0.5 cm) after the visible epimysium was trimmed. Approximately 20 g of these pieces were combined with 40 mL of chilled ultrapure water and homogenized at 4000 rpm for 15 s. The resulting homogenates underwent filtration through 100-mesh sieves to remove the water and the sarcoplasmic proteins, and the residues were washed twice with 1.1 M KCl to obtain the initial connective tissue. Then, the obtained connective tissue was stirred in 20 mL of 1 M KCl solution at 4 °C for 24 h. Subsequently, the crude connective tissue was stirred with 0.9% sodium chloride solution for 24 h. Finally, after two washes in ultrapure water, it was centrifuged to get the IMCT.

### 2.10. Differential Scanning Calorimetry Analysis

DSC analysis was performed using a Perkin-Elmer DSC 8000 (Perkin Elmer Instruments Co., Ltd, Waltham, MA, USA) according to Zhang et al. [28]. Subsequently, the extracted IMCT was sectioned into pieces (weighted 15 mg) for testing. The experiments utilized a heating rate of 10 °C/min, spanning temperatures from 25 to 95 °C. The enthalpy of denaturation (ΔH) was determined by measuring the peak area.

### 2.11. Fourier Transform Infrared Spectroscopy Analysis

FTIR spectra of IMCT were determined by a Nicolet iS10 FTIR spectrometer (Madison, WI, USA) following Han et al.’s [29] method. Spectral data were collected over the range of 4000–400 cm^−1^ with a resolution of 4 cm^−1^ for 64 accumulations. Deconvolution fitting was performed on the amide I and III. bands to determine their relative areas.

### 2.12. Statistical Analysis

Data were subjected to one-way ANOVA followed by Duncan’s multiple range test using SAS version 9.12 (SAS Institute Inc., Cary, NC, USA). Significance was determined at *p* < 0.05, and the results are reported as the mean and standard error of the mean. The results of Pearson correlation coefficients were visualized using a heat map. Principal component analysis (PCA) was performed using SIMCA-P version 11.5 (Umetrics, Sweden).

## 3. Results and Discussion

### 3.1. Macroscopic Appearance and Histology of WB

Representative cases of normal breast fillets and of moderate and severe WB fillets are shown in Figure 1a. Normal breast fillets exhibited consistent softness and flexibility, while WB fillets were palpably tough. Specifically, moderate WB fillets showed hardness in the cranial area but flexibility in the mid to caudal region, whereas severe WB fillets were characterized by extreme rigidity from the cranial region to the caudal tip. In addition, severe WB was accompanied with petechial hemorrhage and edema on the dorsal surface. These morphologic features were in accordance with previous findings [2,30]. As exhibited in Figure 1b, H&E staining showed tightly packed muscle fibers in the normal group, whereas WB samples exhibited aggregation of mononuclear cells, myodegeneration, necrosis, and lipid and connective tissue infiltration (indicated by arrows). Masson Trichrome staining and Sirius red staining demonstrated a significant accumulation of collagen in WB muscles, especially in the endomysial and perimysial spaces, in a similar manner to previous studies [6,7]. In addition, these pathological features became aggravated as the severity of WB myopathy increased.

### 3.2. Chemical Composition

The chemical composition of broiler breast meat was significantly affected by WB abnormality (Table 1). Specifically, compared with the normal group, significant increases in moisture and fat contents were observed in WB fillets (*p* < 0.05). Conversely, WB fillets exhibited lower levels of protein and ash compared with normal breast fillets (*p* < 0.05). Additionally, these distinctions escalated with the heightened severity of WB myopathy (*p* < 0.05). Zhang et al. [31] have also observed that WB meat had higher contents of moisture and fat than those of normal meat, whereas the contents of protein and ash showed the opposite trends. Muscle damage or pathology can induce the disorganization of muscle structure and alterations of muscle fiber characteristics, which inevitably lead to changes in the chemical composition of meat [32]. Therefore, the decrease of protein content and the increase of fat content in WB fillets are probably due to myodegeneration and the infiltration of adipose tissues, as indicated in the H&E staining (Figure 1). The higher moisture content in the WB fillets can be ascribed to the accumulation of ECM glycosaminoglycans as well as the presence of oedema [2,33]. Furthermore, the reduction of ash content in WB fillets might be related to the loss of liquids and the altered ion homeostasis induced by this myopathy [34].

### 3.3. Meat Quality Traits

As depicted in Table 1, meat color showed no significance among the three groups (*p* > 0.05). The pH value of severe WB fillets was higher compared with that of normal breast fillets, aligning with the observations by Tasoniero et al. [35]. Studies have shown that the increase in muscle pH of WB myopathic birds may be attributed to the lower glycogen content and impaired glycolytic capacity [36]. The cooking loss of WB fillets was significantly higher than that of normal breast fillets (14.31 vs. 17.67 and 21.58, respectively; *p* < 0.05). Additionally, cooking loss was observed to increase with the increasing severity of WB myopathy (*p* < 0.05). This may be due to myodegeneration as well as the enhanced depositions of IMCT and adipose tissue, which destroyed the water retention capacity of muscle protein. The integrity of the fiber membrane is impaired by muscle fiber degeneration, resulting in water loss during cooking [37]. Tenderness is regarded as one of the crucial indicators by which consumers can evaluate meat quality. Herein, we observed a higher shear force value of severe WB meat as compared with the normal meat (45.10 vs. 32.78; *p* < 0.05), indicating that WB myopathy resulted in a deterioration of tenderness. Similarly, Chatterjee et al. [38] concluded that the Meullenet–Owens Razor Shear force was higher in WB meat samples compared with normal breast meat, attributing this to the accumulation of interstitial connective tissue.

### 3.4. Textural Properties

Generally, hardness refers to the internal binding force required for food to maintain its shape and cohesiveness and is influenced by the structural integrity of food. Chewiness refers to the amount of work undertaken when food is chewed to the point where it can be swallowed [39]. Distinct variations in hardness, gumminess, and chewiness were observed between normal and WB fillets (Table 1, *p* < 0.05). However, no differences were observed regarding the springiness, cohesiveness, and resilience (*p* > 0.05). Using a similar texture technique, Zhang et al. [31] have also indicated higher values of hardness, gumminess, and chewiness in WB fillets versus normal breast fillets. Moreover, these differences were more pronounced in severe WB than in moderate WB. However, Thanatsang et al. [40] have found that there is no significant difference in hardness between WB and normal meat. Indeed, different instrumental techniques measuring meat texture might result in variable results. It should also be noticed that the effect of WB myopathy on textural characteristics was not uniform throughout the fillet, as Maxwell et al. [41] have indicated that the differences of textural properties caused by WB myopathy are predominantly in the ventral portion of the fillet.

### 3.5. Water Mobility and Distribution

The representative distribution of T_2_ transversal relaxation time identified by LF-NMR in normal and WB muscles is shown in Figure 2a. Severe WB muscles had longer relaxation times of T_21_ and T_22_ and a shorter relaxation time of T_23_, compared with the normal muscles (Figure 2b, *p* < 0.05). WB myopathy decreased the proportions of bound water (P_21_) and immobile water (P_21_), whereas it increased the proportion of free water (P22), which is a similar result as found in our previous findings [23]. These results imply a greater mobility of bound water that is immobile within muscle structure, as well as a leakage of water from intra-myofibrillar spaces into extra-myofibrillar spaces, which reasonably explains the increased cooking loss of WB [42]. The altered water mobility and distribution within WB muscle was associated with the reduction of myofibrillar proteins and the consequent water-binding capacity caused by myodegeneration and insufficient regeneration [43].

### 3.6. Collagen Profile

Collagen is the predominant protein in connective tissue, the solubility of collagen is related to the tenderness of meat [44]. As depicted in Figure 3a, the total collagen content in muscles affected by WB was notably higher compared with that of normal muscles, which is result that is in line with the findings of Zhu et al. [17]. Diffuse interstitial thickening with connective tissue was the main cause of higher total collagen content in WB [6]. No significant difference was observed in the content of soluble collagen (Figure 3b, *p* > 0.05). Nevertheless, WB myopathy significantly decreased collagen solubility (Figure 3c, *p* < 0.05). Collagen attributes, especially content and solubility, determine the contribution of IMCT to meat tenderness. According to a resent meta-analysis, collagen content contributed positively to Warner–Bratzler shear force (WBSF), whereas collagen solubility was negatively related to WBSF [45]. The effect of thermal treatment on meat texture is related to the solubilization of connective tissue collagen [46]. Chen et al. [47] have found that the collagen solubility of the breast meat of spent hen was much lower than that of Arbor Acres broiler, which is in correspondence with the higher WBSF value of spent hen.

### 3.7. Major IMCT Components

The effects of WB myopathy on the contents of cross-links and core proteoglycans in chicken breast muscle are shown in Figure 3c–e. Collagen can obtain additional mechanical properties and chemical stability through intermolecular and intramolecular crosslinking formed after translation [13]. HP is a kind of cross-linking between collagen molecules, which occurs between the terminal peptide of the collagen trimer and a quarter of the staggered adjacent trimer helical region. The current results indicate that WB myopathy significantly increased HP content (*p* < 0.05). This is similar to the findings of Zimmerman et al. [48], who observed an accumulation of collagen with a simultaneous occurrence of increased HP cross-linking in myocardium after acute myocardial infarction. The increase of HP cross-linking makes collagens continuously form a three-dimensional network structure with each other, which further increases meat toughness [49]. Proteoglycans, comprising a diverse family of macromolecules with a central core protein and at least one covalently attached glycosaminoglycan (GAG) chain, play crucial roles in regulating the size and arrangement of connective tissue fibers, influencing meat texture. Moreover, interactions between proteoglycans and collagen and non-collagen components can impact tissue function and structure [50]. The content of GAG in the WB fillets was significantly higher than that in the normal chicken breast (*p* < 0.05). GAGs are generally thought to be involved in the formation of ECM, thereby supporting tissue architecture [51]. Decorin, belonging to the small leucine-rich proteoglycan family, consists of a core protein of around 45 kDa, along with a single covalently attached chondroitin or dermatan sulfate chain. Compared with the normal breast, the content of decorin gradually increased in WB fillets (*p* < 0.05). Velleman and Clark [33] have also found a higher mRNA expression of decorin in the PM muscle of WB myopathic broilers. Decorin is a regulator of collagen cross-linking and fibril structure [52]; a higher level of decorin contributes to tightly packed collagen fibers and the stiffness of WB.

### 3.8. Thermal Properties of IMCT

The denaturation temperature and enthalpy reflect protein thermal stability. Higher values are indicative of better thermal stability [53]. With an increase of the severity of WB myopathy, the thermal transition temperature shifts towards a higher temperature (Figure 4a). The denaturation temperatures of To (onset temperature), T_M_ (maximum temperature) and T_E_ (end-point temperature) of IMCT isolated from severe WB muscle were significantly increased compared with the normal IMCT (*p* < 0.05). While the denaturation enthalpy (ΔH) showed an opposite trend (*p* < 0.05, Figure 4b–e). The thermal stability and mechanical stability of IMCT are primarily due to the chemical properties of collagen molecules [12]. Kopp et al. [54] have found that the ΔH of collagen in IMCT decreases with an increase of collagen crosslinking. Zhu et al. [17] have also indicated that the change of IMCT structure crosslinking in WB is the main reason for the decrease of ΔH. Therefore, the changes of collagen content and the differences in cross-linking of IMCT are the main reasons for the difference of denaturation ΔH between normal and WB chicken muscle.

### 3.9. Secondary Structure of IMCT

There are different vibration modes of protein amide bonds, mainly including amide A, amide B, amide I, amide II, amide III, and amide IV regions (Figure 4f). The amide I band is the C=O stretching vibration peak of the protein polypeptide skeleton and the COO^−^ antisymmetric contraction vibration peak of the α-helix. This is also a sensitive area for analyzing protein secondary structure and is often used for that purpose. Infrared spectra were processed using peak fit 4.12 software, and the amide I band was Gaussian fitted to calculate the subpeak area for determining the relative content of each secondary structure. As exhibited in Figure 4g, the main structure of IMCT isolated from chicken PM muscle is β-sheet (more than 45%). In comparison with the normal group, the content of α-helix and β-sheet in IMCT of WB significantly decreased, and the contents of β-turn and random coil significantly increased (*p* < 0.05). This finding aligns with the results of Zhang et al. [31]. The decline of α-helix content and the increase of random coil content indicate the unfolding of collagen molecules and their stretched molecular structures [55]. An et al. [56] have reported that the secondary structure of surimi protein is altered by the TGase-catalyzed cross-linking reaction, leading to the transition of α-helix and unordered structures to β-sheet and β-turn. Similarly, the changes in the secondary structure of IMCT are speculated to be related to the increased crosslinking caused by WB myopathy.

### 3.10. Correlation Analysis

Correlation coefficients for physiochemical characteristics of IMCT, chemical composition, meat quality traits and textural properties are illustrated in Figure 5a. There were significant correlations (*p* < 0.05) between chemical composition and IMCT composition (total collagen, decorin, HP crosslinking), as well as thermal properties and secondary structure of IMCT. For the water holding capacity (WHC), we observed that cooking loss and the proportion of free water were positively correlated (*p* < 0.05) with total collagen content, HP crosslinking, decorin content, and denaturation temperature of IMCT, and negatively correlated (*p* < 0.05) with collagen solubility and content of α-helix and β-sheet in IMCT. Similarly, Wang et al. [57] have indicated that pork *semitendinosus* muscle had better WHC than *longissimus thoracis* and *semimembranosus* muscles, though they showed lower contents of cross-links and decorin. Shear force and hardness were positively correlated (*p* < 0.05) with total collagen content, HP crosslinking and decorin content, and negatively correlated (*p* < 0.05) with collagen solubility. The relationship between meat tenderness and IMCT characteristics has been well documented. Florek et al. [58] have demonstrated a positive correlation between the shear force of cooked meat and the total collagen content in muscle tissue, as well as a negative correlation with the solubility of collagen. Based on the aforementioned discussions, HP cross-linking is related to the tight parallel packing of collagen, and the major proteoglycan decorin plays vital roles in stabilizing the collagen fibril structure [52,59], which certainly contribute to meat tenderness and toughness. Principle component analysis (PCA) was also conducted and the score plot (Figure 5b) illustrated that the first two principal components explained 50.4% of the total variation (39.9% and 10.5%, respectively). The correlation loading plots showed that the moisture content, cooking loss, shear force and water mobility and distribution had significantly positive relationships with the contents of total collagen, decorin, HP crosslinking, the denaturation temperatures of IMCT and the contents of β-turn and random coil. Collagen solubility showed negative effects on the textural properties. In summary, IMCT correlated well with meat quality attributes, which is mainly manifested in the effect of a series of changes in the characteristics of collagen on tenderness and hardness, involving collagen content, thermal solubility, crosslinking and thermal denaturation. However, the role of IMCT in WHC remains to be elucidated.

## 4. Conclusions

Histological features of myodegeneration, lipid and IMCT accumulation certainly lead to altered chemical composition and impaired meat quality. Particularly, WB myopathy increases the contents of major IMCT components (collagen, HP cross-link, decorin and GAGs) and alters thermal stability and the secondary structure of IMCT, which contribute to the elevated shear force value and hardness of meat. In addition, correlation analysis implied potential contributions of IMCT to other meat quality traits, such as water distribution and WHC, which in turn deserve further investigation.

## Figures and Tables

**Figure 1 foods-13-00507-f001:**
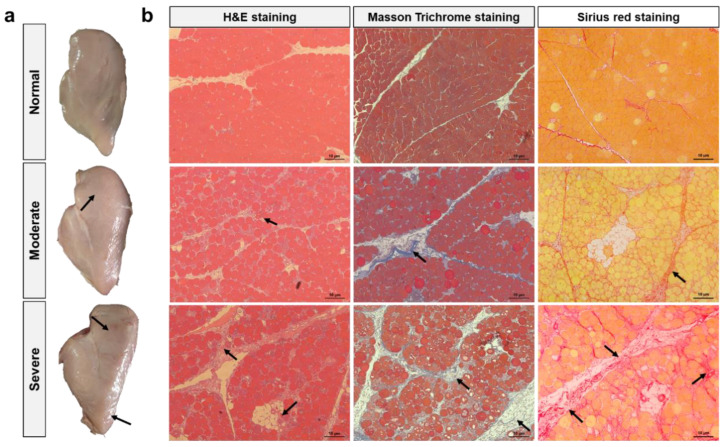
Morphology and histology of PM muscles affected by varying degrees of WB condition. (**a**) Representative photographs showing the morphology of normal, moderate and severe WB fillets. (**b**) Representative images of hematoxylin and eosin staining, Masson Trichrome staining, and Sirius red staining showing the histology of normal, moderate and severe WB muscles.

**Figure 2 foods-13-00507-f002:**
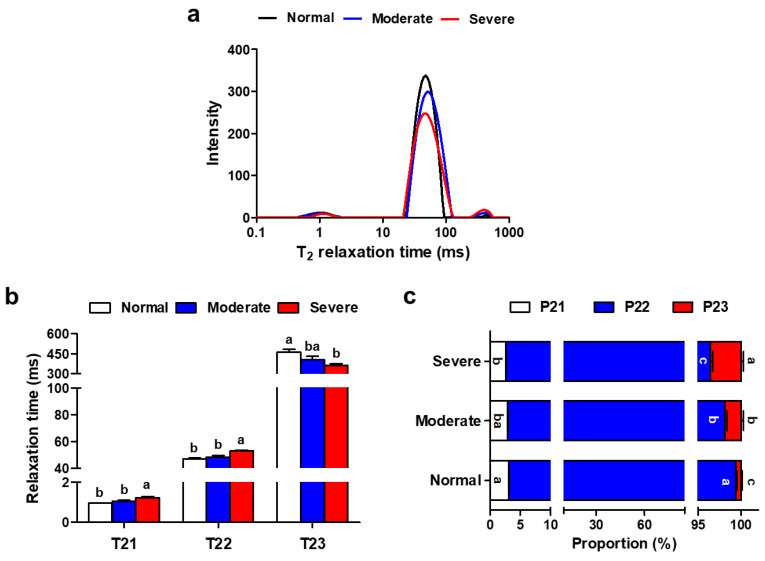
Relaxation properties of pectoralis major muscles affected by varying degrees of WB condition. (**a**) Distributions of transverse relaxation times (T2) for normal, moderate and severe WB muscles. (**b**) Transverse relaxation times (T21, T22 and T23) obtained from normal, moderate and severe WB muscles. (**c**) Relative populations (P21, P22 and P23) obtained from normal, moderate and severe WB muscles. All results are presented as mean ± SE (*n* = 12) with different superscript letters (a–c) denoting significance (*p* < 0.05).

**Figure 3 foods-13-00507-f003:**
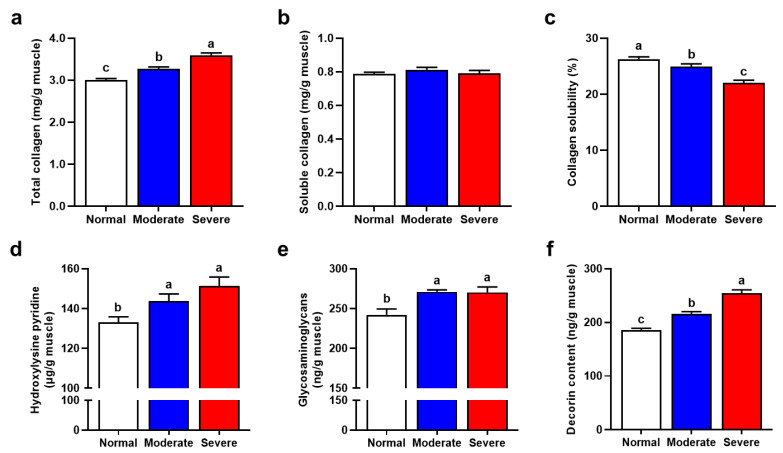
IMCT compositions affected by varying degrees of WB condition. (**a**–**c**) Content of total collagen and heat soluble collagen, as well as the collagen solubility in normal, moderate and severe WB muscles. (**d**–**f**) Content of hydroxylysyl pyridoxine, glycosaminoglycan and decorin in normal, moderate and severe WB muscles. All results are presented as mean ± SE (*n* = 12) with different superscript letters (a–c) denoting significance (*p* < 0.05).

**Figure 4 foods-13-00507-f004:**
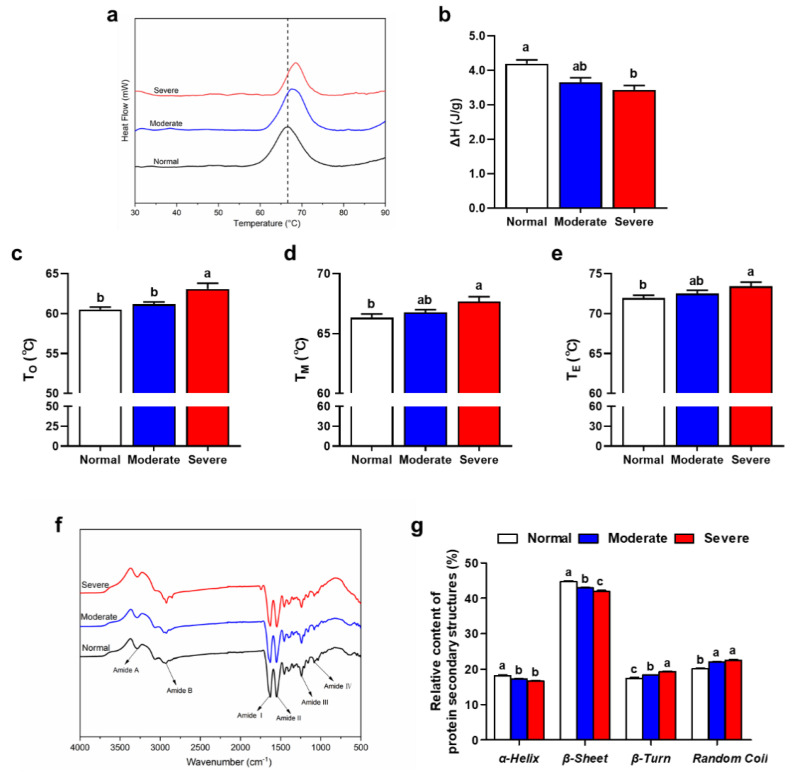
Thermal property and secondary structure of the IMCT affected by varying degrees of WB condition. (**a**) Representative differential scanning calorimetry heat flow diagram of IMCT. (**b**) Enthalpy (ΔH) of IMCT. (**c**–**e**) Maximum transition temperature of IMCT. (**f**) Representative Fourier transform infrared (FT-IR) spectra of IMCT. (**g**) Relative content of protein secondary structures of IMCT. All results are presented as mean ± SE (*n* = 12) with different superscript letters (a–c) denoting significance (*p* < 0.05).

**Figure 5 foods-13-00507-f005:**
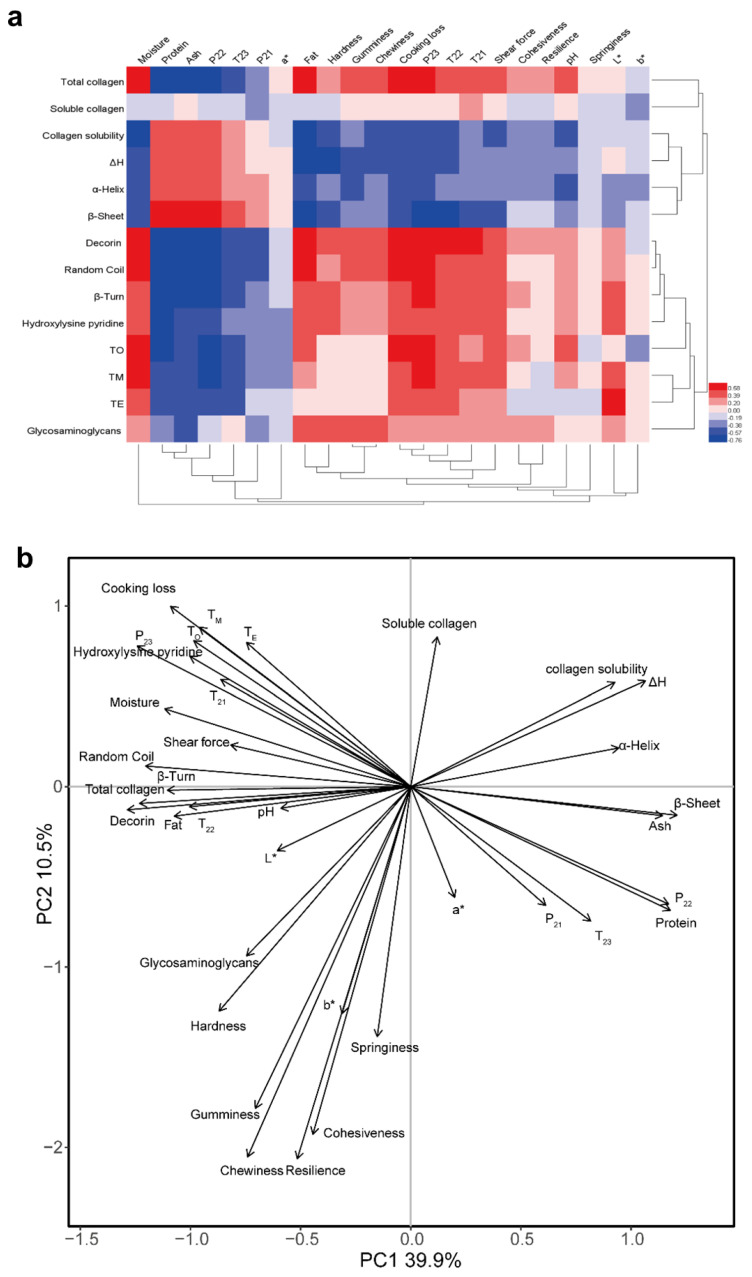
Relationship among physiochemical characteristics of intramuscular connective tissue, chemical composition, meat quality traits and textural properties of pectoralis major muscles affected by varying degrees of wooden breast (WB) condition. (**a**) Heat map illustrating correlation coefficients among variables. Red color indicates positive correlation, whereas blue color indicates negative correlation. (**b**) Loading plot of the first two principal components (PCs) of a principal component analysis.

**Table 1 foods-13-00507-t001:** Effect of WB condition on chemical composition, meat quality attributes and textural properties of broiler breast fillets (*n* = 12).

Items	Category			SEM	*p* Value
	Normal	Moderate	Severe		
Chemical composition					
Moisture (%)	73.29 c	74.13 b	75.26 a	0.24	<0.001
Protein (%)	23.75 a	22.77 b	21.61 c	0.21	<0.001
Fat (%)	1.21 c	1.74 b	2.37 a	0.14	<0.001
Ash (%)	1.47 a	1.39 b	1.32 c	0.01	<0.001
Meat quality parameters					
Lightness	51.72	52.63	53.38	0.7	ns
Redness	4.44	3.85	4.3	0.36	ns
Yellowness	6.72	6.91	6.81	0.42	ns
pH	5.91 b	5.94 ba	6.00 a	0.03	0.047
Cooking loss (%)	14.31 c	17.67 b	21.58 a	0.49	<0.001
Shear force (N)	32.78 b	36.37 b	45.10 a	2.36	0.003
Textural properties					
Hardness (g)	2814.96 b	3164.68 ba	3576.05 a	180.6	0.019
Springiness (mm)	0.55	0.55	0.56	0.02	ns
Cohesiveness	0.48	0.48	0.52	0.02	ns
Gumminess (g)	1308.92 b	1807.60 a	1860.76 a	155.7	0.029
Chewiness (g·mm)	739.69 b	1009.53 ba	1087.39 a	101.3	0.04
Resilience	0.22	0.24	0.25	0.03	ns

a–c Means within rows with different lowercase letters differ significantly (*p* < 0.05). ‘ns’ denotes not significant.

## Data Availability

Data is contained within the article.

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
