# Peer review of "Effects of Wooden Breast Myopathy on Meat Quality Characteristics of Broiler Pectoralis Major Muscle and Its Changes with Intramuscular Connective Tissue"

_foods, 2024, doi:10.3390/foods13040507_

Round 1

Reviewer 1 Report

Comments and Suggestions for Authors

Manuscript  ID: foods-2858221 Relationship between physiochemical characteristics of intramuscular connective tissue and meat quality of chicken pectoralis major muscle affected by wooden breast myopathy

General opinion

Thank you for entrusting me with reviewing your manuscript. The authors undertook research related to the influence of physiochemical characteristics of intramuscular connective tissue on the meat quality of chicken pectoralis major muscle affected by wooden breast myopathy. I find the manuscript very interesting. The research carried out is very up-to-date due to the increased incidence of abnormalities in the breast muscles of chickens. The authors performed many analyzes and obtained results confirming the influence of wooden breast myopathy on the physicochemical properties of intramuscular connective tissue and its contribution to the variability of meat quality. The obtained results may have an impact on improving the quality of products from chicken meat affected by myopathy. The manuscript is clearly written, but the texture profile analysis method (section 2.5) should be described in more detail. For example, there is no information at what speed the test was performed and what was the relaxation time between compressions. Texture parameters should also be explained (section 3.4.) and the units of texture parameters should be provided in Table 1.

Author Response

Reviewer 1#

Comment: The authors undertook research related to the influence of physiochemical characteristics of intramuscular connective tissue on the meat quality of chicken pectoralis major muscle affected by wooden breast myopathy. I find the manuscript very interesting. The research carried out is very up-to-date due to the increased incidence of abnormalities in the breast muscles of chickens. The authors performed many analyzes and obtained results confirming the influence of wooden breast myopathy on the physicochemical properties of intramuscular connective tissue and its contribution to the variability of meat quality. The obtained results may have an impact on improving the quality of products from chicken meat affected by myopathy. The manuscript is clearly written, but the texture profile analysis method (section 2.5) should be described in more detail. For example, there is no information at what speed the test was performed and what was the relaxation time between compressions.

Response: Thanks for your recognition of our work. We have described the method of texture profile analysis (section 2.5) in more detail according to your comment (Line 115-117 of revised manuscript).

Comment: Texture parameters should also be explained (section 3.4.) and the units of texture parameters should be provided in Table 1.

Response: Thanks for this comment. We have explained and discussed the changes of texture parameters (Line 239-243 of revised manuscript). In addition, the units of some texture parameters have been in revised Table 1.

Reviewer 2 Report

Comments and Suggestions for Authors

Dear Authors, please find my comments below as it follows:

The title of the article should be rephrased. It seems that you compared physicochemical characteristics with meat quality?...Physicochemical characteristics are, or could be considered as a „part“ of meat quality properties…and when reading your aim it seems that you only investigated the composition and physicochemical properties of IMCT. It is a little confused.

Is it known/hypothesized why WB occurs? Could you please incorporate the information related to this into introduction?

L73 – which commercial broiler line?

L74 – which standard?

L 88-90- under which conditions and magnification?

L 210 – please report more in detail which results were reported by Zhang et al.

Figure 5 – should be more clear. It's really hard to focus on some details..without reading discussion, challenging

Conclusion – please conclude, do not repeat your results. 

Author Response

Reviewer 2#

Comment: The title of the article should be rephrased. It seems that you compared physicochemical characteristics with meat quality?...Physicochemical characteristics are, or could be considered as a „part“ of meat quality properties…and when reading your aim it seems that you only investigated the composition and physicochemical properties of IMCT. It is a little confused.

Response: Thanks for this comment. We have rephrased the title in Line 2-4 of revised manuscript. In addition, we also revised the aim of this study in the Abstract section to meet with your suggestion (Line 13-15 of revised manuscript).

Comment: Is it known/hypothesized why WB occurs? Could you please incorporate the information related to this into introduction?

Response: Thanks for this comment. To date, extensive studies have been conducted to depict the causes of this rapid growth-related broiler breast muscle myopathy. Although the precise etiology of WB myopathy remains not fully understood, accumulating evidence has shown that localized tissue hypoxia, oxidative stress, mito-chondrial dysfunction and glucolipotoxicity are involved in the development of WB. We have incorporated some information into the introduction section (Line 41-44 of revised manuscript).

Comment: L73 – which commercial broiler line?

Response: Thanks for this comment. We have incorporated this information in Line 74 of revised manuscript.

Comment: L74 – which standard?

Response: Thanks for this comment. We have incorporated this information in Line 75-77 of revised manuscript.

Comment: L 88-90- under which conditions and magnification?

Response: Thanks for this comment. We have incorporated the detailed description of image acquisition conditions in Line 90-91 of revised manuscript.

Comment: L 210 – please report more in detail which results were reported by Zhang et al.

Response: Thanks for this comment. We have described the results of Zhang et al. in Line 203-205 of revised manuscript.

Comment: Figure 5 – should be more clear. It's really hard to focus on some details without reading discussion, challenging

Response: Thanks for this comment. We have re-uploaded the figures for a clear exhibition (Revised Figure 5).

Comment: Conclusion – please conclude, do not repeat your results. 

Response: Thanks for this comment. We have rephrased the conclusion in Line 399-405 of revised manuscript.